# FROM PIXELS TO PROSE:
# A LARGE DATASET OF DENSE IMAGE CAPTIONS

## ABSTRACT

Training large vision-language models requires extensive, detailed image-text pairs. Existing web-scraped datasets, however, are noisy and lack detailed image descriptions. To bridge this gap, we introduce PixelProse, a comprehensive dataset of over 16M (million) synthetically generated captions, leveraging cutting-edge vision-language models for detailed and accurate descriptions. To ensure data integrity, we rigorously analyze our dataset for problematic content, including child sexual abuse material (CSAM), personally identifiable information (PII), and toxicity. We also provide valuable metadata such as watermark presence and aesthetic scores, aiding in further dataset filtering. We hope PixelProse will be a valuable resource for future vision-language research. PixelProse will be made available publicly.

## 1 INTRODUCTION

Early vision-language models were trained on datasets of images from the web, each labeled with the alt-text embedded in the surrounding HTML. These datasets enabled model training at large scales for numerous applications. However, as models advanced and the machine learning community moved, these datasets have begun to outlive their usefulness. The problems with these datasets stem from the fact that alt-texts are not truly captions. They often contain little to no information about the content of the image, and factors like background objects and fine-grained details are often absent. As a result, commercial models that are trained on purpose-labeled and carefully curated datasets have *far* surpassed the open source state of the art for both image generation and analysis. Overall, trending research in the community has shown that dataset quality, not dataset size, has become the bottleneck for open-source development. This motivates the need for new datasets that are labeled with deliberately constructed captions rather than incidental alt-texts. At the same time, the emergence of generative LLMs enables fast manipulation and reformatting of text labels. This raises the value of *dense* image labels containing many categories of detailed information, as one dataset can be refactored for many uses including vision captioning and question-answering (VQA).

PixelProse is a dataset that addresses the weaknesses of existing alt-text datasets for vision-language applications and is designed to be used as either a standalone asset or in combination with LLM refactoring. It contains detailed captions that are long, detailed, and cover a range of image properties that are important for Vision-Language Model (VLM) and diffusion model training, as depicted in Figure 1. Rather than target only one specific application (e.g., VQA), PixelProse captions are intended to be *general purpose* image descriptions that contain large amounts of image data in dense prose form. These captions can be used for pre-training tasks, image captioning, or they can be refactored into other data formats (e.g., VQA, instructions, etc.) using an LLM.

## 2 PIXELPROSE DATASET

In this section, we provide a detailed description of how we created the PixelProse dataset. An overview of our data generation pipeline is shown in Figure 2. Our captions are generated using Google Gemini 1.0 Pro Vision Model (Team et al., 2023). The images from the dataset are provided as URLs, along with original and generated captions.

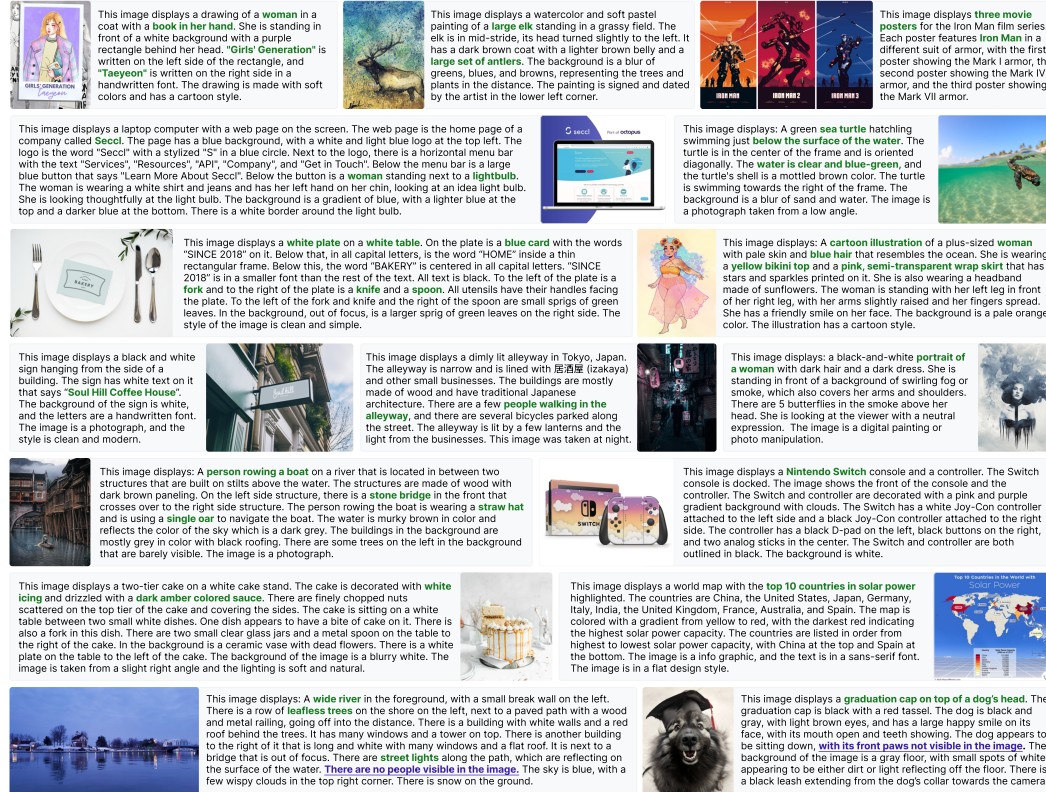

Figure 1: Dense synthetic image captions from PixelProse. Concrete phrases are highlighted in green, and negative descriptions are underlined in purple.

## 2.1 IMAGE SOURCES

PixelProse comprises over 16M diverse images sourced from three different web-scraped databases, which are discussed below:

**CommonPool** (Gadre et al., 2024) contains a large pool of image-text pairs from CommonCrawl, which is distributed as a list of url-text pairs under a CC-BY-4.0 License. We filter the dataset using cld3[1] to detect English-only text and select image-text pairs with a CLIP-L/14 similarity score above 0.3. This filtering scheme is the same as LAION-2B (Schuhmann et al., 2022), and is supported through the metadata provided with the dataset. From our filtered subset, we recaption over 6.2M samples.

**CC12M** (Changpinyo et al., 2021) comprises 12.4M web-crawled images and alt-text pairs. The dataset is curated using both image and text-based filters. From this dataset, we recaption over 9.1M samples.

**RedCaps** (Desai et al., 2021) is curated from Reddit. It consists of 12M image-text pairs from 350 different subreddits, which are filtered to select general photographs and minimize the number of people (such as celebrity images). The images are fairly high quality, while captions are non-descriptive. From this dataset, we sample and recaption nearly 1.2M samples.

Our goal in choosing data sources is to achieve a wide range of image properties and quality/aesthetic rankings. The CommonPool data is less strictly curated than other sources, contributing lower quality images, (which are important for VLM training) and high diversity. Also, it is collected more recently and contributes more current information about celebrities and locations. The CC12M dataset features higher image quality and is subject to stricter curation. The RedCaps images are the most strictly curated by humans and are of very high quality and artistic value on average.

---

[1] https://github.com/google/cld3

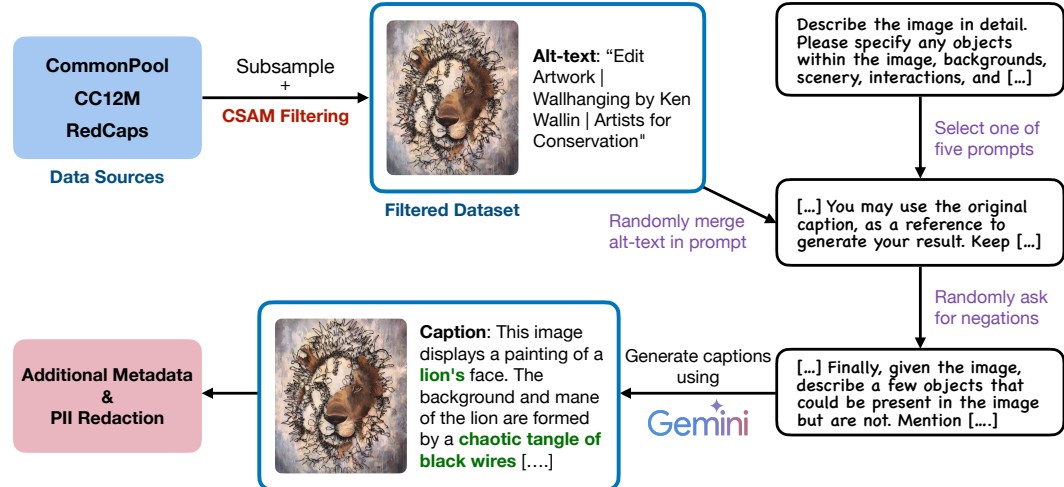

Figure 2: Illustration of our pipeline for generating detailed, and diverse synthetic captions. We sample image and alt-text pairs from various sources while filtering for CSAM content. We adopt our strategy to generate prompts that are then used to produce captions with Google Gemini 1.0 Pro Vision Model (Team et al., 2023). Finally, we redact different forms of PII and provide additional metadata such as aesthetic scores.

## 2.2 TEXT CAPTIONING

We aim to generate detailed image descriptions containing types, attributes, and counts of objects in an image, in addition to spatial relations between objects, the presence of text, various broad image categorizations, etc.

**Prompting Strategy**. We use five unique prompts to diversify the generated captions. Each asks for descriptions with various attributes. These prompts are provided in A.1. We showcase one of the prompts used below.

---

```
Describe every component of this image, as it were described by an artist in at
most two paragraphs. Each object, with its count, positions, and attributes
should be described. Describe the text, and the font in detail with its
contents in quotation marks. For example if the image has text Happy Birthday,
write it down as "Happy Birthday". Include the style of the image for example
photograph, 3d-render, shopping website etc. Capture the aesthetics of the
image, as if described by an artist. Start with the words 'This image
displays:'
```

---

In addition to selecting one of the five prompts, we randomly also add a reference to the original alt-text pair within the prompt. Prior work (Yu et al., 2024) has found this strategy helps improve descriptive accuracy when alt-texts contain useful information, particularly proper nouns (e.g. "Taj Mahal" instead of "White Marble Mausoleum").

**Negative Descriptions**. Despite their impressive capabilities, both text-to-image diffusion models and VLMs exhibit weaknesses in understanding negative instructions. For example, telling a diffusion model to create an image with "no elephant" is likely to create an image with an elephant, while asking a VLM about an elephant when there is none is likely to produce a hallucination. Such poor behaviors probably arise in part because online image captions seldom deliberately reference absent objects.

To foster a better language understanding of negative references, we also prompt Gemini to describe absent objects for a subset of images. We manually verify that prompting helps generate meaningful negative captions, as depicted in Figure 1. Depending on the application, these negatives can easily be filtered out based on the metadata or the final sentences in the generated caption.

**Text Recognition**. Reading or generating text in images is essential for VLMs and diffusion models. To support this, PixelProse features a substantial caption component that identifies text within images. To ensure text recognition accuracy, we manually spot-check images and their corresponding captions. First, we classify images using our watermark model (Section 3.3) and identify images without a watermark but with the text present. Then, we apply an OCR spotting model (Baek et al., 2019) to these images. We discard images with text regions smaller than 15 pixels in width or height.

Finally, we did a manual assessment to confirm text recognition accuracy in our captions. We attempted to automate this study using OCR classification models for text recognition and caption overlap checks, but found that inaccuracies due to fragmented text regions and OCR errors made this infeasible. The results of our manual study are presented in Table 1. For roughly 76% of the images, the text within the captions is correctly recognized.

Table 1: Spot-check results (image ratio percentage) for text recognition in 100 image captions.

| Correct | Incorrect | Not Captured |
|---------|-----------|--------------|
| 76%     | 4%        | 20%          |

However, text recognition in captions fails in challenging cases, such as highly arbitrary or rotated shapes and highly artistic fonts. We discuss some of these examples in the Appendix A.2.

## 2.3 ETHICAL CONSIDERATIONS

A growing body of work discusses potential ethical concerns regarding data scraped from the internet (Birhane et al., 2024; Birhane & Prabhu, 2021; Gebru et al., 2021). Several large-scale datasets used for training machine learning systems have come under scrutiny, prompting a reevaluation and in some cases withdrawal of these datasets (Birhane & Prabhu, 2021; Yang et al., 2022; Asano et al., 2021; Thiel, 2023). These datasets have been misused for various applications. For example, text-to-image generative models trained on large-scale datasets can generate NSFW content resembling specific individuals. Birhane et al. (Birhane et al., 2024) found that LAION-2B (Schuhmann et al., 2021) contains hate content, highlighting problems of uncurated large-scale datasets.

### 2.3.1 NSFW & CSAM FILTERING

Recent work has shown that text-to-image models are trained on and can even produce Child Sexual Abuse Material (CSAM) content (Thiel et al., 2023; Thiel, 2023). In a recent study, LAION-5B (Schuhmann et al., 2022) was found to contain CSAM and subsequently taken down [2] (Thiel, 2023). Addressing CSAM in future datasets requires robust detection mechanisms and better data collection practices [3]. We discuss our approach to removing CSAM, and other NSFW content below.

First, the image sources for our dataset already employ different mechanisms to remove NSFW content. The CC12M (Changpinyo et al., 2021) dataset was filtered using commercial Google APIs for detecting pornographic and profane content in both images and alt-text descriptions. RedCaps (Desai et al., 2021) removed any subreddits or posts marked as NSFW (either by authors or subreddit moderators). They further used an open-source NSFW classification model [4] to filter the remaining content. CommonPool (Gadre et al., 2024) uses a modified version of LAION-5B (Schuhmann et al., 2022) CLIP-based NSFW classification model. The classifier was further validated against Google Vision API's SafeSearch explicit content detector.

To further ensure the safety and integrity of our data, we check our dataset against several commercial APIs. First, we use the PhotoDNA API by Microsoft [5], which uses perceptual hashing to match against a database of known CSAM content. PhotoDNA is regarded as the industry standard and can detect such content even if the images are slightly altered (Farid, 2021). We specifically process the images we sampled from the CommonPool dataset against the PhotoDNA API, as our other data sources are already processed to filter CSAM using different industrial APIs (Iwatt et al.; goo). Finally, all our data is processed through Google Gemini API (Team et al., 2023) which provides additional safeguards. The API blocks prompts (including images) and responses against certain core harms such as child safety [6]. We found 92 matches against the PhotoDNA database, all of which were

---

[2] https://laion.ai/notes/laion-maintenance/
[3] https://info.thorn.org/hubfs/thorn-safety-by-design-for-generative-AI.pdf
[4] https://github.com/GantMan/nsfw_model
[5] https://www.microsoft.com/en-us/PhotoDNA
[6] https://ai.google.dev/gemini-api/docs/safety-settings

removed from PixelProse. One should not conclude that our original data sources contain CSAM, as these examples were not flagged by the Google Gemini API and were likely to be false positives.

### 2.3.2 PERSONALLY IDENTIFIABLE INFORMATION (PII)

Recent works have highlighted the use of PII in large datasets (Koh et al., 2024; Lukas et al., 2023). To ensure privacy, PII redaction steps are integrated into our data processing pipeline. We remove images, and captions from PixelProse that contain phone numbers. We found no Social Security Numbers (SSNs) in the captions. Phone numbers and SSNs are detected and redacted using regular expressions that search for various standard PII number formats (e.g., (123)-456-7890,

Table 2: PII comparison between the original and PixelProse captions. The values represent the percentage of captions containing names, phone numbers, E-mail IDs, and SSNs.

|  | Names | Phone Numbers | E-mail IDs | SSNs |
|---|---|---|---|---|
| Original Captions | 10.51% | 0.05% | 0.32% | 0.00% |
| PixelProse | 7.93% | 0.12% | 1.23% | 0.00% |

123-456-7890, and 123.456.7890). We additionally run the *anonymization* and *scrubadub* Python packages over image captions as an additional filter, to ensure that PII is removed.

We find that our generated captions contain more phone numbers and e-mail IDs than the original captions. This indicates that our dataset contains rich labels of text content, but also highlights the need for robust PII scrubbing mechanisms to protect sensitive information.

Table 3: Toxicity level comparison between the original and PixelProse captions using Detoxify (Hanu & Unitary team, 2020) at a threshold of 0.2. The values represent the percentage of captions exhibiting each type of toxicity. PixelProse captions show significantly lower toxicity scores across all attributes, indicating improved safety and content quality.

|  | Threshold | Toxicity | Severe Toxicity | Obscene | Identity Attack | Insult | Threat | Sexual Explicit | Overall Toxicity |
|---|---|---|---|---|---|---|---|---|---|
| Original Captions | 0.2 | 0.74% | 0.00% | 0.08% | 0.07% | 0.26% | 0.04% | 0.04% | 0.75% |
| PixelProse | 0.2 | 0.13% | 0.00% | 0.03% | 0.00% | 0.06% | 0.00% | 0.01% | 0.13% |

### 2.3.3 TOXICITY

Mitigating toxicity in datasets is vital for ethical AI deployment. Previous research (Deshpande et al., 2023; Zhuo et al., 2023; Wen et al., 2023; Gehman et al., 2020) highlights that language models are prone to various forms of toxicity, such as hate speech, identity hate, explicit content, insults, and harmful stereotypes. To address these concerns, we conduct a toxicity analysis of our generated captions using Detoxify (Hanu & Unitary team, 2020), which classifies text across a wide range of toxic attributes, from overtly offensive language to subtle passive-aggressive remarks. We subsequently flag $0.13\%$ of captions using a threshold of $0.2$ across all attributes.

Our analysis in Table 3 shows that the PixelProse captions are safer compared to the original captions. Most of our captions fall within the lowest toxicity range (0-0.2) across various attributes. Specifically, the percentages of captions exhibiting severe toxicity, identity attacks, and threats are exceptionally low, with PixelProse achieving $< 0.01\%$ for all three. For overall toxicity, PixelProse captions exhibit a markedly lower percentage of $0.13\%$ compared to $0.75\%$ for the original captions. For this reason, we believe PixelProse is well suited for training generative models with low risk of harmful outputs.

## 3 A CLOSER LOOK AT THE DATASET

### 3.1 LINGUISTIC DIVERSITY

In Figure 3, we show the distribution of caption lengths for our generated captions compared to the original caption. The generated captions are generally more descriptive and contain more words. Our generated captions average 506 characters per caption, compared to 101 characters for the original captions, and are longer for over $98\%$ of the data. Figure 4 shows the histogram of the number of tokens based on LLaMA-3 (lla, 2024) tokenizer. PixelProse comprises 1,710,499,128 (1.7B) text tokens.

In Table 4, we show the noun diversity across several open-source datasets recaptioned using different captioning models (Bird et al., 2009). Our dataset offers a larger noun vocabulary across images

compared to other datasets. Our dataset is two orders of magnitude larger than ALLaVA (Chen et al., 2024a), and an order of magnitude larger than ShareGPT4V (Chen et al., 2023). SAM-LLaVA (Chen et al., 2024c) of similar scale to our dataset, but is captioned using the LLaVA-1.0 7B model (Liu et al., 2023b) that suffers from significant hallucinations (Chen et al., 2024b; 2023).

Table 4: We analyzed the noun vocabulary in multiple datasets recaptioned using different models, defining valid nouns as those that appear more than 10 times. We found that PixelProse is larger and has a more diverse noun vocabulary than other datasets. Our dataset is also complementary to other datasets in that it covers different sources of images, and was captioned by a different commercial model.

| | Size | Image Sources | Captioning Model | Valid Nouns | Distinct Nouns | Total Nouns |
|---|---|---|---|---|---|---|
| ALLaVA Chen et al. (2024a) | 0.68M | VisionFlan, LAION | GPT-4V(ision) | 18K | 121K | 23.32M |
| ShareGPT4V Chen et al. (2023) | 1.34M | CC3M, SBU, LAION, etc. | Multiple | 13K | 66K | 49.26M |
| SAM-LLaVA Chen et al. (2024c) | 11.5M | SAM | LLaVA-1.0 | 23K | 124K | 327.90M |
| Ours | **16.4M** | CC12M, RedCaps, etc. | Gemini 1.0 Pro | **49K** | **490K** | **357.61M** |

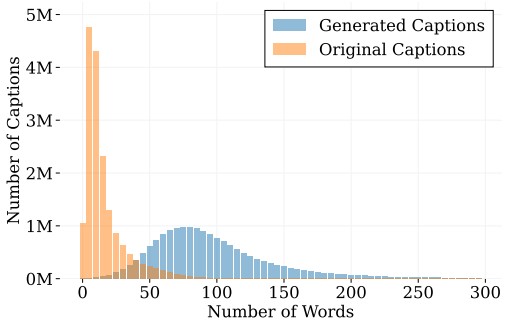

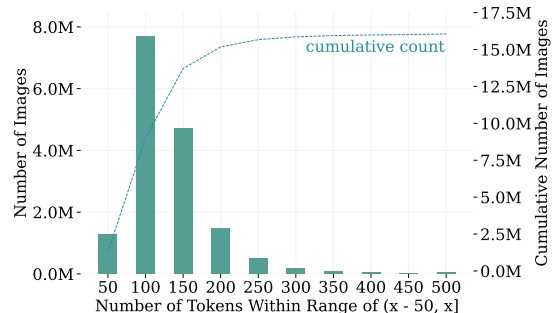

Figure 3: Histogram of words for generated captions v/s original captions. Generated captions are longer with an average of 106 words, while original captions only have 19 words on average.

Figure 4: Histogram of tokens for generated captions, which are tokenized by the tokenizer of LLaMA-3 (lla, 2024). The bars at 150 represent the number of images with (100, 150] tokens in their captions.

## 3.2 REPURPOSING CAPTIONS INTO VQA PAIRS

Our captions contain dense general-purpose information and are intended to be ideal inputs for LLM refactoring. To probe how our captions can be refactored into specific formats, we use LLaMA-3 8B Instruct (lla, 2024) to refactor our captions into free-form VQA pairs for 100 images. We manually verified that over 70% of the VQA pairs generated using our captions were valid pairs. Figure 5, shows some of these VQA Pairs. Other works have shown refactoring captions into VQA pairs or other instructions can be further improved using better language models and prompting strategies (Liu et al., 2023b). We discuss the details of refactoring captions into VQA pairs in the Appendix A.3.

## 3.3 STATISTICS OF PIXELPROSE CONTENT

We quantitatively describe the PixelProse dataset by reporting the size, watermark prevalence, aesthetic scores, and style attributes of images.

**Image Resolution**. In PixelProse, over 15M images have a resolution below 2000 pixels, while the rest are high-resolution images exceeding 2000 pixels, as shown in Figure 6. For each data source, the average sizes are as follows: (299.6, 331.9) ± (137.2, 149.5) for CommonPool, (719.7, 820.0) ± (269.0, 285.1) for CC12M, and (1234.1, 1234.4) ± (277.2, 325.2) for RedCaps.

**Watermark Detection**. To detect and label the presence of explicitly visible watermarks in images, we follow the work of (Schuhmann et al., 2022).[7] However, we found that this method leads to frequent false-positives in the case that images are without watermark but with innocuous text. This is problematic, as an explicit goal of our efforts is to include and properly label images containing

---

[7] https://github.com/LAION-AI/LAION-5B-WatermarkDetection

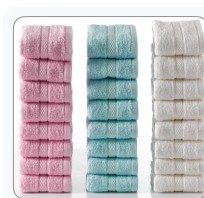

**Question:** What are the colors of the towels in each stack?
**Answer:** The towels are **pink, blue, and white**, respectively.

**Original Caption:** Perfect 15 Incredible Small Bathroom Decorating Ideas

**Our Caption:** This image displays three stacks of folded hand towels. [.....] The stacks are arranged in a row, with the **pink** towels on the left, the **blue** towels in the middle, and the **white** towels on the right. There is a white background and the towels are stacked vertically. The image is a photograph.

**Question:** How many beer taps are there on top of the fridge?
**Answer:** There are **two beer taps** on top of the fridge.

**Original Caption:** Hands on Review: KOMOS Stainless Steel Kegerators! - Designed for Home brewers

**Our Caption:** This image displays a stainless steel mini fridge with **two beer taps** on top of it. There is a black drip tray under the taps. The fridge has a black handle and a digital display on the front. There is a brick wall in the background. The image is well-lit and the fridge is the main focus.

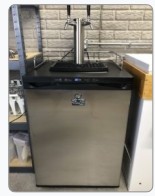

Figure 5: Our captions are much more detailed than the original alt-text pairs, and can be refactored into VQA Pairs. We use our detailed captions to prompt Llama3-8B Instruct, a text-only model to generate question/answer pairs. The images are shown only for reference.

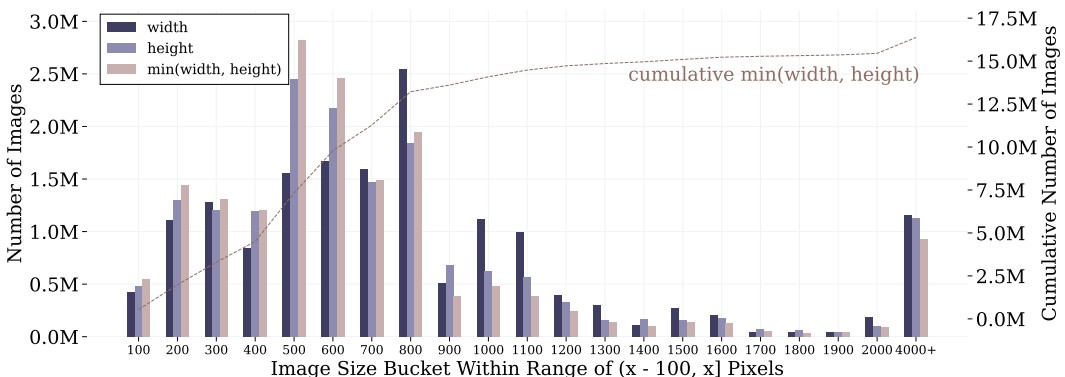

Figure 6: Histogram of image size. Each bucket is within the range of $(x - 100, x]$ pixels, e.g., bars at 700 represent the image count with $(600, 700]$ pixels. The bin at $4000+$ considers $(2000, 4000+]$.

text. To mitigate this, we manually collected an additional group of hard examples to fine-tune the model. These images fell into three categories: with watermark, without watermark, and without watermark but with text, as demonstrated in Figure 7.

The figure also demonstrates the corresponding probability score for each category. To better understand the distribution of images w/ or w/o watermark in the whole dataset, we plot the histogram for the three categories within different score ranges in Figure 8. The lowest probability scores for all three categories are around 0.5. We carefully review images with low probability scores around 0.5 in the two watermark-free categories, noting that they are still safe to keep.

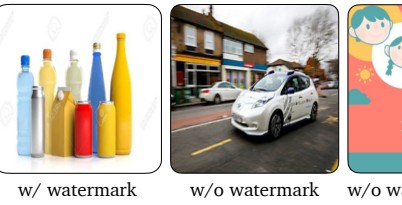
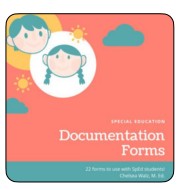

w/ watermark
score: 0.93

w/o watermark
score: 0.96

w/o watermark but
w/ text (score: 0.96)

Figure 7: Categories of watermark classification models.

For the watermark category, we recommend a filtering threshold above 0.85, indicating that less than 6% of the dataset (around 1M images) are truly watermarked in PixelProse.

**Aesthetic Estimation.** Aesthetically pleasing images tend to have clearer and more distinct visual features, which may help in learning better representations for VLMs. This is also crucial for diffusion models to generate high-quality, visually appealing images. Most importantly, aesthetic images often have more coherent and contextually relevant descriptions, aiding in better alignment between images and captions.

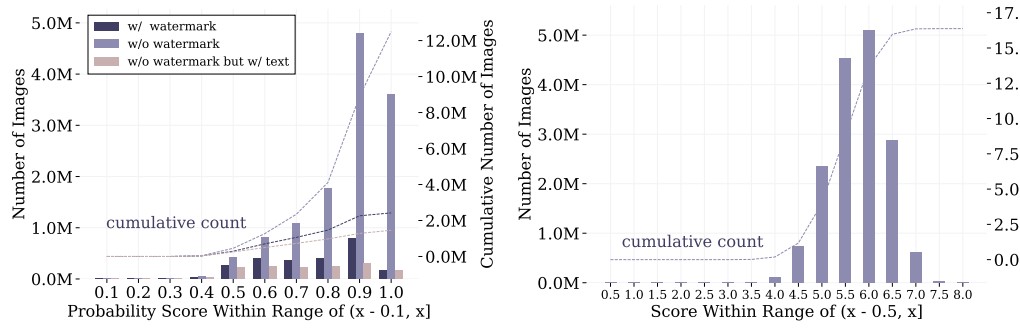

Figure 8: Histogram of watermark scores, with each bucket in the range $(x-0.1, x]$, e.g., bars at 0.7 indicate the image count with scores between $(0.6, 0.7]$.

Figure 9: Histogram of aesthetic scores, with each bucket in the range of $(x-0.5, x]$. For example, bars at 7.0 indicate the image count with scores between $(6.5, 7.0]$.

To investigate aesthetic properties in PixelProse, we fine-tune the aesthetic filter LAION-Aesthetics V2 (Schuhmann et al., 2022) with natural and generated (synthetic) images selected from recent high-quality datasets (Xu et al., 2023; Yi et al., 2023; Huang et al., 2008; Karras et al., 2018; Kirstain et al., 2023). We semi-manually annotate our filtered training data, giving higher scores to more artistic, realistic, high-definition, and text-based data sources. To supervise training, we adopt the mean value of the original aesthetic predictor and our annotations as the label.

Figure 9 shows the distribution of images based on their aesthetic scores. Images with scores below 5.0 generally are blurry or less artistic (see Figure 10) and make up a small portion of PixelProse compared to those with relatively high scores. These images are still valuable for augmenting training due to the diversity they bring to the overall dataset. Most images have relatively high aesthetic scores above 5.0, indicating PixelProse contains a large proportion of high-quality images (more than 11M).

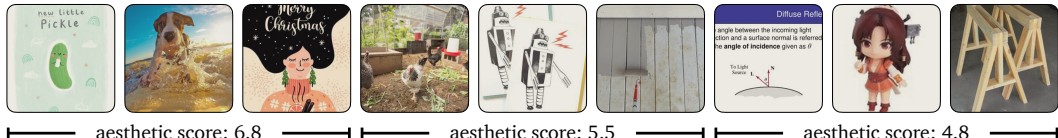

Figure 10: Images with corresponding aesthetic scores.

## 4 EXPERIMENTS

To understand how PixelProse dataset, can be useful for vision language applications, we perform experiments by pretraining and finetuning vision-language models using our dataset. We show improved performance across different benchmarks.

For our finetuning experiments, we use a pretrained PaliGemma model[8] (Beyer et al., 2024). We compare against 100K GPT4V captions from ShareGPT4V dataset, and original raw captions for PixelProse Chen et al. (2023). We experiment with 2M randomly sampled captions, from our dataset. To control for the dataset size, we also compare against using 100K samples from the CommonPool subset of our data. For evaluation, we focus on several popular benchmarks, such as VQVA2 (Goyal et al., 2017) for visual question answering, OCRBench (Liu et al., 2024c) for optical character recognition, NoCaps (Agrawal et al., 2019) for novel object captioning, and DetailCaps Dong et al. (2024) for detailed object captioning.

The results are shown in Table 5. Note that ShareGPT4V contains a total of 100K captions from a GPT4V, while our dataset contains 16M captions. PaliGemma fine-tuned on PixelProse 2M synthetic captions outperforms other datasets on nearly all evaluations. Specifically, on visual question answering, and OCR models trained on our PixelProse synthetic captions perform better than ShareGPT4V and PixelProse using original captions.

---

[8]https://huggingface.co/google/paligemma-3b-pt-224

| Finetune Dataset | VQVA2 / Lite Accuracy | OCRBench Score | NoCaps BLEU_1 | NoCaps ROUGE | NoCaps METEOR | DetailCaps / All CAPTURE | DetailCaps / Gemini CAPTURE |
|---|---|---|---|---|---|---|---|
| None | 45.42 | 289 | 0.08 | 0.24 | 0.105 | - | - |
| GPT4V 100K | 58.38 | 419 | 0.313 | 0.258 | 0.205 | **0.593** | 0.4355 |
| PixelProse Original 2M | 56.2 | 339 | 0.25 | 0.231 | 0.097 | - | - |
| PixelProse Ours 100K | 57.3 | 342 | 0.345 | 0.284 | 0.233 | 0.566 | 0.544 |
| PixelProse Ours 2M | **59.88** | **455** | **0.344** | **0.291** | **0.238** | 0.5611 | **0.5481** |

Table 5: We finetune PaliGemma on different vision-language datasets, and evaluate their performance across multiple benchmarks for visual-question answering, optical character recognition, and image captioning.

| Pretrain Dataset | FineTune Dataset | Accuracy |
|---|---|---|
| ShareGPT4V 1.2M | VQA-V2 Train | 55.37 |
| PixelProse Original 3M | VQA-V2 Train | 60.8 |
| PixelProse Ours 3M | VQA-V2 Train | 68.44 |

Table 6: Results for pre-training with different vision language datasets, then finetuning and evaluation for visual-question answering. The model performs best when pretrained on PixelProse synthetic captions.

For DetailCaps, we observe that the model finetuned on ShareGPT4V performs slightly better. However, DetailCaps contains ground-truth captions from three different models (GPT4o, GPT4V and Gemini-Pro 1.5). As it contains two GPT models as ground-truth, it may skew evaluations in favor of GPT4V data (i.e ShareGPT4V). Hence, we also report results using only Gemini Pro 1.5 as the ground-truth in the last column. Here, we again observe that the models trained on PixelProse captions perform better than ShareGPT4V by a larger margin.

When training using only 100K samples from PixelProse, we again outperform GPT4V on NoCaps, and Details (Gemini). On OCRBench, our performance is worse. However, we note that ShareGPT4V carefully curated data specifically for text due to its smaller size. Our dataset is much larger, and leveraging 2M samples we are able to outperform GPT4V using 100K samples.

We also conduct experiment with pre-training vision language models. We use all the ShareGPT-4V 1.2M images (100K GPT4V images, and 1.1M ShareCaptioner images), and 3M images from CC12M subset of PixelProse dataset. We use CLIP-L/14 image encoder Radford et al. (2021), and Gemma 2B language model Gemma Team (2024). We first pre-train a small MLP as our multi-modal adapter using PixelProse original and synthetic captions. For the first stage, we only train the adapter and do not fine-tune the vision / language models. Then, we fine-tune on VQAV2 training split and evaluate on the the full VQAV2 validation split. Our results are shown in Table 6, showing for pretraining, our synthetic captions again outperform ShareGPT4V and PixelProse original captions.

## 5 RELATED WORK

Many large-scale image caption datasets such as COYO-700M (Byeon et al., 2022), DataComp (Gadre et al., 2024), LAION (Schuhmann et al., 2021), YFCC100M (Thomee et al., 2016), CC12M (Chang-pinyo et al., 2021), SBU (Ordonez et al., 2011), RedCaps (Desai et al., 2021) are created from various internet sources by mapping an image to its corresponding alt-text or the text surrounding the image. Despite their large sizes, the quality of captions for these datasets is quite low.

Higher quality image caption datasets such as MS-COCO (Lin et al., 2014), VizWiz (Gurari et al., 2020), VisualGenome (Krishna et al., 2017), nocaps (Agrawal et al., 2019), Flickr30K (Young et al., 2014), TextCaps (Sidorov et al., 2020) and many others (Pont-Tuset et al., 2020; Kazemzadeh et al., 2014; Mao et al., 2016) exist, however, they are usually smaller (sub-million) in size. The LLaVA (Liu et al., 2024a; 2023a;b; 2024b) family of models and a series of smaller VLMs (Li et al., 2024; Chu et al., 2024) have shown that it is possible to train a high-performance model with small-scale synthetic data (Chen et al., 2023) from GPT-4(V). PixArt-$\alpha$ (Chen et al., 2024c) trained a higher-quality diffusion model with 25M images, and VLM caption pairs with approximately 1.25% training data volume compared to Stable Diffusion v1.5 (Rombach et al., 2022). Stable Diffusion v3 (Esser et al., 2024) also uses 50% VLM synthetic captions for training diffusion models. Many other recent works (Zhou et al., 2023; Chen et al., 2024c; Kondratyuk et al., 2024; Chen et al., 2024b; Liu et al., 2024a) have also shown that a few million higher quality examples can train better models than

many million low-quality data. Hence, high-quality datasets are urgently needed to train the next generation of multi-modal models.

A few attempts are made towards this goal, completely human-annotated, with humans-in-the-loop, or some completely automated. DOCCI (Onoe et al., 2024) is a small high-detailed image caption dataset that is completely human-annotated. Despite having only 15K samples, all the captions contain diverse details like key objects and their attributes, spatial relationships, text rendering, and so on. ImageInWords (Garg et al., 2024) is another small-scale detailed caption dataset that takes a slightly different approach by using object detection and other annotation models with humans in the loop. Densely Caption Images (DCI) (Urbanek et al., 2023) is another human-in-the-loop annotation dataset which uses labels from Segment Anything (Kirillov et al., 2023). Both these datasets contain fewer than 10K samples.

LVIS-Instruct4V (Wang et al., 2023) dataset contains detailed captions of 110K images from the LVIS (Gupta et al., 2019) dataset annotated by GPT-4V (Achiam et al., 2023). ALLaVA (Chen et al., 2024a) introduces 715K captions by GPT-4V on images sourced from LAION (Schuhmann et al., 2021) and Vision-Flan (Xu et al., 2024). ShareGPT4V dataset contains 100K detailed captions on images sourced from LAION (Schuhmann et al., 2022), SBU (Ordonez et al., 2011), and CC12M (Changpinyo et al., 2021) created by GPT-4V. They further train a model and generate captions for over a million images. LLaVA (Liu et al., 2023a) introduces a dataset of 23K detailed captions on top of COCO images using GPT-4. Lastly, Pixart-$\alpha$ (Chen et al., 2024c) introduces large-scale synthetic captions on top of the SAM dataset (Kirillov et al., 2023) using LLaVA-1.0 7B (Liu et al., 2023a) model. While this particular dataset contains 11M examples, it contains many captions with hallucinations and the images in the dataset are of limited diversity. PixelProse has over 16M samples, which to the best of our knowledge is the largest detailed high-quality publicly available image-caption dataset.

## 6 LIMITATIONS AND CONCLUSION

Our images are collected from the internet, which contains unsafe and toxic content. Though we use extensive automated measures to remove CSAM, NSFW content, and PII, our automated systems are imperfect. VLMs tend to suffer from hallucinations, hence the captions may not always accurately describe the image. While we use a state-of-the-art large commercial model to generate our captions, it still suffers from hallucinations. Despite this, our captions are of *much* higher quality and fidelity than captions in other similar-sized public datasets. Most importantly, unlike the original alt-text captions, PixelProse captions consistently reflect the image content.

In addition to its obvious uses in training open-source models, we hope that the dense format of Pixel-Prose facilitates research into methods for refactoring dense captions into instructions and VQA pairs.

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

# APPENDIX

## A.1 PROMPTS

We utilize five different prompts for our dataset, which are provided below. Some of these prompts were taken and adapted from other sources such as LAION-Pop [9].

---

Describe the image in detail. Please specify any objects within the image, backgrounds, scenery, interactions, and gestures or poses. If they are multiple of any object, please specify how many and where they are. If any text is present in the image, mention where it is, and the font.Describe the text in detail with quotation marks. For example, if the image has text, Merry Christmas, write it down as "Merry Christmas". Describe the style of the image. If there are people or characters in the image, what emotions are they conveying? Identify the style of the image, and describe it as well. Please keep your descriptions factual and terse but complete. The description should be purely factual, with no subjective speculation. Make sure to include the style of the image, for example cartoon, photograph, 3d render etc. Start with the words 'This image displays:'

Describe every component of this image, as it were described by an artist in atmost two paragraphs. Each object, with its count, positions, and attributes should be described. Describe the text, and the font in detail with its contents in quotation marks. For example if the image has text Happy Birthday, write it down as "Happy Birthday". Include the style of the image for example photograph, 3d-render, shopping website etc. Capture the aesthetics of the image, as if described by an artist. Start with the words 'This image displays:'

Describe the image, the foreground and the background. All objects, along with its count and positions must be described. For any text present in the image, describe the text using quotation marks. Be factual in your description, capturing the content, and style of the image. Describe the image, in a short but desciptive manner. Start with the words 'This image displays:'

Write a detailed caption describing the image. Include all components, and objects with their positions. If any text is present in the image, and describe the text contents in quotation marks. For example if the image has text Happy Birthday, write it down as "Happy Birthday". Be detailed in your description of the image, and write as if it were being described by a boring person. Start with the words 'This image displays:'

Don't forget these rules: 1. Be Direct and Concise: Provide straightforward descriptions without adding interpretative or speculative elements. 2. Use Segmented Details: Break down details about different elements of an image into distinct sentences, focusing on one aspect at a time. 3. Maintain a Descriptive Focus: Prioritize purely visible elements of the image, avoiding conclusions or inferences. 4. Follow a Logical Structure: Begin with the central figure or subject and expand outward, detailing its appearance before addressing the surrounding setting. 5. Avoid Juxtaposition: Do not use comparison or contrast language; keep the description purely factual. 6. Incorporate Specificity: Mention age, gender, race, and specific brands or notable features when present, and clearly identify the medium if it's discernible. When writing descriptions, prioritize clarity and direct observation over embellishment or interpretation. Write a detailed description of this image, do not forget about the texts on it if they exist. Also, do not forget to mention the type/style of the image. No bullet points. Start with the words, "This image displays:"

---

[9] https://laion.ai/blog/laion-pop/

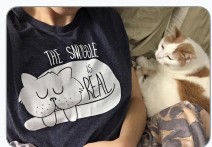

**Our Caption:** This image displays a young woman wearing a gray t-shirt with the text "**The snuggle is real**" and a picture of a sleeping cat on it. The woman is lying in bed with the cat, who is curled up next to her. The cat is white and orange, and has one paw on the woman's arm. The woman is smiling down at the cat. The background is a white wall with a window. The text on the shirt is in a handwritten font. The image is a photograph. The emotion conveyed is happiness.

**Our Caption:** This image displays a wall with two framed black-and-white photographs of watches on it. There is also a small red, white, and blue flag with a single star in the upper left corner of the flag. The flag has gold fringe and is attached to the wall by a small nail. There is a black hat on top of the framed photos. There is text at the top of the photo that reads: "**GENESTONE AND STEPHEN PULVIRENT**".

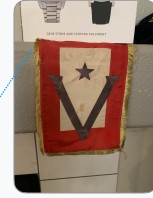

Figure A.1: Images with their corresponding captions. The correctly recognized text is highlighted in green.

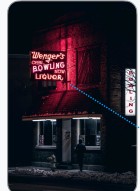

**Our Caption:** This image displays a night scene of a retro **bowling** galley with a **liquor** sign. A man with dark hair and a black jacket walks toward the entrance. The bowling alley has red neon signage and a brick exterior. Its windows are covered with snow. A fire hydrant is located on the sidewalk in front of the entrance. The image iclear and well-lit.

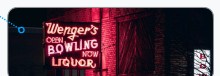

**Our Caption:** This image displays a book titled "**The Prevent and Reverse Heart Disease Cookbook**", which is signd by two people. The book is open and there is a handwritten note that is partially visible. The note says "**To Everette**, **Anne**, **You know the power**. **Thrive**. **A**nn Crile Esselstyn, **M.D.**" The background of the image is black, and the book is white with blue text.

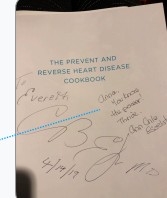

Figure A.2: Images with imperfect text recognition in the captions. The correctly recognized text is highlighted in green and incorrect text is highlighted in red.

## A.2 TEXT RECOGNITION

We observe that captions fail to capture text in images when text data is in a complex format, or the model fails to adhere to the prompt. Failure cases for text recognition are shown in Figure A.2. Despite some failure cases, text recognition is fairly successful. We show several cases in Figure A.1.

## A.3 VQA CONSTRUCTION

To construct our VQA pairs using caption data, we use LLaMa-3-8B Instruct a text-only model lla (2024). We use the following user prompt to construct our VQA pairs.

```
You are an AI visual assistant, and you are seeing a single image. What you see
are provided with is context regarding the image, describing the same image
you are looking at. Answer all questions as you are seeing the image. Design a
conversation between you and a person asking about this photo. The answers
should be in a tone that a visual AI assistant  is seeing the image and
answering the question. Ask diverse questions and give corresponding answers.
Include questions asking about the visual content of the image, including the
object types, counting the objects, object actions, object locations,
relative positions between objects, etc. Only include questions that have
definite answers: (1) one can see the content in the image that the question
asks about and can answer confidently; (2) one can determine confidently from
the image that it is not in the image. Do not ask any question that cannot be
answered confidently. Here is the image description:
```

Since vision-language models tend to hallucinate, several VQA pairs are invalid however based on
our manual spot check, we find that over 70% of our constructed VQA pairs are valid.

### A.4   IMAGE STYLE ATTRIBUTES

Style attributes play a key role in organizing, retrieving, analyzing, and personalizing image content.
They enhance the usability of the dataset, making them more valuable for various applications. For
example, categorizing images based on style simplifies the retrieval of specific types of images from
our large dataset. If a user is searching for documentary chart images, having this as a category
enables quick estimation of the number of available images and ensures accurate retrieval.

As shown in the example prompt in Section 2.2, Gemini is tasked with providing the style of the
image in its response. These responses are then analyzed and categorized to a predefined set of classes
based on the occurrence of specific keywords, as listed in Table A.1. Table A.2 offers an insight into
the relative frequencies of image style categories across PixelProse, showing that photographs are the
most prevalent image style within our dataset, followed by painting, drawings, comics and digital art.

Table A.1: Predefined Vocabulary for Image Style Categorization. The category "other" includes medical images,
screenshots, and captions that do not fit into the existing categories.

| Image Type Category | Sample Keywords |
| --- | --- |
| Photographs | photograph |
| 3D Rendering | render, 3D, 3d, 3-dimensional |
| Digital Art | digital, CGI, CG, vector, raster |
| Painting and Drawings | paint, draw, sketch, comic, anime |
| Charts & Diagrams | chart, plot, diagram, table, map |

Table A.2: Distribution of image type categories across PixelProse. Gemini responses are analyzed, and each is
assigned to a category based on the occurrence of a predefined set of words in the style part of the caption.

| Image Type | Photographs | Painting and Drawings | 3D Rendering | Digital Art | Chart or Diagrams | Other |
| --- | --- | --- | --- | --- | --- | --- |
| Relative Frequency | 85.9 | 4.3 | 3.5 | 1.0 | 0.5 | 4.8 |

### A.5   BROADER IMPACTS

Internet data can reflect societal biases, which already exist in our data sources, i.e., CC12M,
CommmonPool, and RedCaps. Thus, our dataset may inherit these biases. We have taken steps to
mitigate these biases by filtering out captions that contain toxic content, as described in Section 2.
Also, it is challenging to ensure the accuracy and reliability of the captions produced by a state-of-the-
art commercial model, which also may contain biases and generate inexistent or incorrect information.
These issues warrant further research and consideration when training upon our dataset to evaluate
models.

