# OpenReview forum: "From Pixels to Prose: A Large Dataset of Dense Image Captions"
_ICLR.cc/2025/Conference — ICLR 2025 Conference Withdrawn Submission_

### Official Review · Reviewer_quuU · 2024-10-22

**Soundness:** 2
**Presentation:** 3
**Contribution:** 2
**Rating:** 5
**Confidence:** 4

**Summary:**

The authors propose a dataset of image-dense caption pairs. The dense captions are obtained using a state of the art generative model (Gemini). The authors then show that training on a subset of this dataset (size 2M) improves one model performance when compared to using the original subset (without synthetic captions) on various tasks.

**Strengths:**

- The writing is clear.
- Great care was taken to filter the dataset from NSFW content as well as personally identifiable content.

**Weaknesses:**

- Comparison of unique nouns is only provided with respect to other datasets of smaller size. However, the proposed innovation in this work is the generative caption enhancement. Therefore, a more apt comparison is between the original dataset and the enhanced one. This is especially the case since the authors claim that alt-text is a limited source of data.
- Results are only provided over a 2M subset of the data. This is a very limited comparison because it is possible that gains are only realized on smaller dataset sizes (2M compared to 16M).  Therefore, to effectively validate the dataset efficacy, the authors should provide a curve over different sizes of PixelProse. This curve would answer whether the benefit from augmenting the dataset captions would diminish with scale.
- Only one model is tested on the proposed dataset. This is quite limited. What is the rationale behind this choice?
- Overall, the novelty of this work is quite limited. It seems like the work is just a bigger version of prior work ShareGPT4V. This is further undermined by the limited experimental results as outlined in weaknesses points 2 and 3.

**Questions:**

Please refer to the weaknesses section. Overall, the scientific soundness of this work is not good enough given points 2 and 3 in the weaknesses section. This is further undermined by the limited novelty given point 4 in the weaknesses section.

---

### Official Review · Reviewer_9jpw · 2024-10-30

**Soundness:** 2
**Presentation:** 3
**Contribution:** 2
**Rating:** 3
**Confidence:** 4

**Summary:**

The paper addresses a problem with existing image-text multimodal datasets -- they are often noisy and their captions are concise, oversimplified and focused toward the main object in the image. This, in result, hinders the performance of models trained with them. To tackle this, the authors propose a recaptioning framework that generates longer and more detailed captions for existing datasets. The authors have taken into consideration multiple ethical issues, filtering unwanted assets. Moreover, the authors thoroughly study the statistics of the proposed dataset and compare it to other dataset.

**Strengths:**

+ The paper is well written and easy to follow. The authors have done a good job in providing motivation for their paper.

+ The consideration of ethical aspects is important, especially on large-scale datasets.

+ The analysis of the dataset statistics and comparison to other datasets is insightful.

**Weaknesses:**

- Novelty - Recaptioning image-text datasets is not new with several such works [1,2,3]. Unfortunately, I do not see significant improvements or new techniques compared to such works.

- Negative description - the authors have claim that TTI and VLMs struggle with understanding negation and thus they include such phrases in the dataset, however, the effect of this is not demonstrated.

- Repurposing captions into VQA pairs - while the motivation for doing so is clear, the authors do not show any empirical evidence of the advantage of doing so with their dataset.

- Experimental section - Training a model with the same amount of data leads to lower performance than training on the GPT4V 100K samples on VQAv2 and OCRBench which are very important benchmarks. While increasing the scale of the dataset leads to improvement, this cannot be attributed solely to the proposed framework as one could generate more captions for the baseline. In addition, the comparison in table 6 was not done with the same amount of data, leading to an unfair comparison.

[1] Improving clip training with language rewrites

[2] FuseCap: Leveraging Large Language Models for Enriched Fused Image Captions

[3] From scarcity to efficiency: Improving clip training via visual-enriched captions

**Questions:**

Please see the weaknesses

---

### Official Review · Reviewer_wP8G · 2024-11-04

**Soundness:** 3
**Presentation:** 3
**Contribution:** 2
**Rating:** 5
**Confidence:** 5

**Summary:**

The paper proposes a new dataset of 16 million parallel image-text pairs consisting of Internet images and automatically generated captions. The paper details the specific strategy to curate the set of images and the strategy to automatically obtain captions from Google Gemini Pro, including detailed prompts and goals in the design of the prompts. The paper validates the extent to which the dataset is useful by finetuning on captioning and visual question answering tasks, and OCR, as well as using it as a source of pretraining for visual question answering.

**Strengths:**

+ A new dataset with 16 million image-text pairs, potentially useful for finetuning models pre-trained on larger but noisier and lower quality data.
+ Extra safety precautions in the collection of images compared to similar datasets and considerations with respect to privacy and toxicity.
+ Summarizes well the literature on the state of image-caption paired datasets and identifies convincingly some gaps and opportunities for improvement in current practices.

**Weaknesses:**

* The dataset is automatically generated, it doesn't seem like it would be difficult to reproduce collecting a similar dataset with the characteristics described here.
* There is some empirical contribution in the construction of the dataset but there is not one clearly identifiable contribution. Looking at the experiments, it is hard to guess what were the factors that contributed to the improved performance. There are no ablations that can show any insights about this.
* Results: GPT4V-100k obtains 59.38 on VQAv2 while the proposed PixelProse-2M dataset obtains 59.88 on the same benchmark. For a 20x scale up, this seems rather a modest improvement. More critically, it is hard to pinpoint what were the design decisions that lead to this improvement.
* The results are arguably insufficient in determining the impact of the proposed dataset. I have some questions about Table 6 but in general, none of the experiments seem to be validating the full extent of the proposed PixelProse-16M dataset, and particularly the experiment for Table 6 seems somewhat arbitrary and does not contextualize the work with respect to the state-of-the-art.

**Questions:**

* For Table 6, what was the purpose on using only the 3M images from the CC12M subset of PixelProse? Also for this table, is the CLIP-L/14 image encoder trained from random weights? What about the Gemma--2B LM, is it also trained from random weights? Because pre-training seems to imply from random weights but this does not seem clear.

**Details Of Ethics Concerns:**

* Many images in this dataset are likely copyrighted. 6% still have watermarks according to the paper. However I don't want to hold this against this paper too much unless the conference chairs have taken a stance against this practice. In other words, the paper does not do anything that has not been common practice so far. My technical and contribution concerns about the paper and my score does not take into account the issue outlined here.

---

### Official Review · Reviewer_CxrS · 2024-11-08

**Soundness:** 3
**Presentation:** 2
**Contribution:** 2
**Rating:** 3
**Confidence:** 5

**Summary:**

The paper introduces PixelProse, a synthetic dataset of over 16M high-quality image captions designed to advance vision-language model training by addressing the limitations of existing web-scraped datasets, which are often noisy and lack detailed descriptions. The dataset also includes rigorous content screening for issues like CSAM, PII, and toxicity, and valuable data including watermark and aesthetic scores.

**Strengths:**

- The paper highlights a critical issue in current vision-language datasets—data quality. By addressing the noisiness and lack of detail in traditional datasets,  this work aims to provide a more reliable data source for training.
- To enhance data quality, the authors employ diverse prompting strategies to capture varied image details, include negative descriptions to explicitly identify absent objects, and integrate OCR to accurately capture and enhance text elements within images.
- To ensure ethical integrity and high quality, the methodology employs automated filtering mechanisms (e.g., Microsoft PhotoDNA API, PII redaction tools) to screen sensitive content and incorporates tools like the LAION-Aesthetics model to generate metadata, providing detailed image information for quality control and customization.

**Weaknesses:**

- The primary issue with this paper is the insufficient experiments. The authors only evaluate their dataset by fine-tuning a pre-trained PaliGemma model, using a randomly selected subset of 2M samples. This experimental setup introduces significant bias, making it difficult to effectively demonstrate the dataset's validity, generalizability, and scalability. To strengthen the evaluation, additional experiments are necessary, including the following:
  - Training on a broader range of multimodal models, such as CLIP, BLIP, LLaVA, and Qwen-VL, across different model sizes. Since this dataset is densely annotated, pre-training and instruction-tuning on various multimodal models would help establish the effectiveness of dense captions.
  - Comparative analysis with existing datasets, including CC3M, CC12M, LAION-400M, LAION-5B, and DataComp-1B, to validate the dataset's performance and added value.
  - Further validation on additional downstream tasks is necessary, including MMU, DocVQA, TextVQA, ChartQA, OCRBench, MMBench, VCR, HallBench, MathVision, etc.
  - Testing model performance by training with different proportions of the dataset, which would provide insights into the dataset's scalability and adherence to scaling laws.

**Questions:**

Refer to weakness.

---

### Official Review · Reviewer_UKww · 2024-11-08

**Soundness:** 3
**Presentation:** 3
**Contribution:** 2
**Rating:** 5
**Confidence:** 4

**Summary:**

The paper proposes a dataset, PixelProse, of images captioned with detailed captions using Gemini Pro. The dataset combines images of three public web-crawled datasets, namely CC12M, RedCaps, and CommonPool, which are filtered with proper safeguards (CSAM and NSFW), English-only captions, and partially based on image-text alignment (clipscore). This results in a total of 16M images, which are recaptioned with Gemini Pro. The captions are descriptive and optionally contain a negative caption coupled with the correct one. PixelProse is compared to other AI-recaptioned datasets and to its original captions. First, the comparison focuses on safety, showing that the new captions have lower toxicity and personal information, and are safe wrt CSAM, NTSW. Then, the comparison focuses on data statistics of Pixelprose, showing higher linguistic diversity (nouns) than others AI-generated datasets, longer captions than original captions, and statistics about image resolution and aesthetic. Moreover, a small study on how to repurpose the dataset as VQA is showed. Finally, the comparison focuses on the performance of vision-language models fine-tuned or trained on PixelProse versus other datasets.

**Strengths:**

1. Despite the AI-generated captions, Pixelprose captions are well-curated through a pipeline which is quite robust and safe
2. A remarkable effort toward providing higher quality captions for web-crawled datasets to the community.

**Weaknesses:**

1. The dataset contribution is made less valuable by the fact that there is no major use-case showing significant improvements. In more details, the authors propose an empirical comparison by fine-tuning Paligemma, or pre-training at small scale CLIP-Gemma adapters. While the results are positives on these two use-cases, both cases assume models to be pre-trained on other (usually proprietary) datasets, which are much larger than the 16M of PixelProse. The provided evidence only partially supports the main claim: _"These captions can be used for pre-training tasks, image captioning, or they can be refactored into other data formats (e.g., VQA, instructions, etc.) using an LLM."_ Moreover, despite the predisposition to VQA fine-tuning given the dataset size and type of captions, the few VQA repurposing examples presented in the paper are not enough to reach any conclusion on the quality of the dataset for VQA fine-tuning.
2. The dataset propose highly detailed captions, however for certain tasks, e.g., localized image-text alignment, is necessary/preferred to bind part of the captions to regions of the image. Here, the localization is missing, hindering this kind of applications.
3. When inspecting the nouns vocabulary (Table 4) there is no comparison versus the vocabulary of original captions. This datapoint would be relevant to understand whether the vocabulary is not shrink. Is there any reason this datapoint was not included?

**Questions:**

Please comment on the weaknesses. Overall, the paper provides some contribution to the community, however, to me, there is not enough empirical evidence that support the quality/need of this dataset for the research community. Recaptioning dataset is a best practice already present in many state-of-the-art pipelines, so the question I still cannot answer after reading the paper is whether this re-captioning is much better than others to deserve CVPR presentation. For now I will go with a borderline reject, but I am happy to increase if the authors can make a stronger case for PixelProse.

---

### Note · Authors · 2024-12-09

I have read and agree with the venue's withdrawal policy on behalf of myself and my co-authors.